# Artificial nighttime lighting impacts visual ecology links between flowers, pollinators and predators

Emmanuelle S. Briolat [1], Kevin J. Gaston [2], Jonathan Bennie [2], Emma J. Rosenfeld[2] & Jolyon Troscianko [1✉]

The nighttime environment is being altered rapidly over large areas worldwide through introduction of artificial lighting, from streetlights and other sources. This is predicted to impact the visual ecology of many organisms, affecting both their intra- and interspecific interactions. Here, we show the effects of different artificial light sources on multiple aspects of hawkmoth visual ecology, including their perception of floral signals for pollination, the potential for intraspecific sexual signalling, and the effectiveness of their visual defences against avian predators. Light sources fall into three broad categories: some that prevent use of chromatic signals for these behaviours, others that more closely mimic natural lighting conditions, and, finally, types whose effects vary with light intensity and signal colour. We find that Phosphor Converted (PC) amber LED lighting – often suggested to be less harmful to nocturnal insects – falls into this third disruptive group, with unpredictable consequences for insect visual ecology depending on distance from the light source and the colour of the objects viewed. The diversity of impacts of artificial lighting on hawkmoth visual ecology alone argues for a nuanced approach to outdoor lighting in environmentally sensitive areas, employing intensities and spectra designed to limit those effects of most significant concern.

[1] Centre for Ecology & Conservation, University of Exeter, Penryn, UK. [2] Environment & Sustainability Institute, University of Exeter, Penryn, UK. ✉email: jt@jolyon.co.uk

A rtificial nighttime lighting has brought great benefits to human societies, however, it also profoundly alters the timing, intensity and spectrum of natural light regimes[1,2]. Associated with human settlement, transport networks and industry, nearly a quarter of the global land area already lies under artificially light polluted nighttime skies[3]. While the area experiencing direct light emissions is harder to estimate, both these extents are growing rapidly[4]. The spectrum of this lighting is also changing, as previously often narrow spectrum lamps (e.g. low pressure sodium) are widely replaced with broad white light-emitting diode (LED) lamps[5].

Artificial light at night has both been predicted and empirically determined to have a wide diversity of biological impacts[6], including on organismal physiology[2,7] and behaviour[8,9], the abundance and distribution of species[10,11], and the structure and functioning of communities and ecosystems[12]. It is also predicted to have profound impacts on the visual ecology of many organisms, and as a consequence both their intraspecific (e.g. sexual signalling[13]) and interspecific interactions (e.g. pollination[14,15]; anti-predator defences[16,17]). These interactions have often co-evolved for millions of years either to enhance the colours for signalling to a specific receiver, or to alter or mask an organism's appearance from predators. In each case the colours have evolved against specific backgrounds to be viewed by the intended visual system under natural lighting (i.e. sunlight, moonlight or starlight[17]). The emission spectra of artificial light sources differ from those that occur naturally, and could thus dramatically alter the visibility of these co-evolved coloration strategies[14].

To date, studies of the impacts of artificial nighttime lighting on visual ecology have focussed almost exclusively on matching the spectral output of different kinds of lamps, or that of the skyglow they give rise to, with indices of the action spectra of species' visual systems[18–21]. However, these offer limited insight into how different artificial lighting types will alter visual ecology in specific systems because they do not account for the critical interactions between emission spectra, surface reflectance, and the receiver's sensitivity to light at different quantal intensities and wavelengths. Under typical daytime light levels the ability of an animal to distinguish any two colours is largely dependent on the signal-to-noise ratios caused by neural pathways[22]. But, under lower light levels photon shot noise limits colour discrimination[14,23,24]. The point at which colour vision becomes limited by light levels varies substantially between species, and typical artificial light from streetlights creates a mosaic habitat with enormous spatial variation in light intensities. Understanding this interaction is critical if knowledge of the impacts on visual ecology is to help shape discussions as to how to design outdoor artificial lighting schemes that minimise adverse environmental effects.

Here we use the nocturnal elephant hawkmoth (*Deilephila elpenor*) visual system as a model to test the impacts of different light types on the visual ecology of these ecologically important insects. In particular, nocturnal moths are highly effective pollinators: they not only transport pollen further than their diurnal counterparts, but each visit is also more likely to result in successful pollination, making their services highly valuable to wildflowers[15,25]. Moths such as nocturnal hawkmoths (Sphingidae) have low-light colour vision which allows them to locate flowers even under starlight levels of illumination[14,23,26], meaning much of their natural range will be subject to relevant levels of visible light pollution, and they are also found in built-up areas near streetlights. In this study, we assess how different artificial light emission spectra will affect the ability of nocturnal hawkmoths to perform visually-driven behaviours, including flower selection for pollination, as well as intra-specific communication and anti-predator behaviours.

## Results

**Hawkmoth perception of floral signals**. We measured the reflectance spectra of the petals and leaves of wildflowers known or thought to be pollinated by hawkmoths (14 species, Supplementary Table 3), and the ability of hawkmoths to detect these flowers against background foliage (the plant's own leaves and grasses) was modelled under a wide range of light sources (Supplementary Table 1) at intensities spanning the range found in areas illuminated by streetlights. Throughout, the performance of artificial lights was compared primarily to full moonlight conditions (Fig. 1). We also tested a range of other reference illuminants appropriate for a range of natural light levels (including twilight, moonlight at different phases and starlight/skyglow); comparisons between artificial lights and these light sources broadly matched the results of comparisons with full moonlight (Supplementary Fig. 9). First, the photon cone-catch quanta were estimated for petals and leaves[14], photon shot noise was then modelled[24], and the coordinates of these measurements were calculated in a modified version of the receptor noise limited (RNL) chromaticity space[27]; (see supplementary methods for modelling details). In terms of their effect on hawkmoth colour perception, artificial lights fell into one of three categories: some enabled the moth to perceive colour contrasts as well or better than under moonlight, others blocked their colour vision, and finally the effects of some lights varied depending on the light intensity and colours viewed (Figs. 1A and 2).

Overall, white LEDs and mercury vapour lights perform similarly to moonlight, or may even enhance chromatic contrasts between flowers and green backgrounds, as perceived by hawkmoths, regardless of flower colour (Fig. 2; Supplementary Fig. 6). By contrast, narrow-band orange LEDs and low-pressure sodium lamps prevent their use of colour vision at all light levels, always yielding chromatic contrasts between flowers and leaves below a Delta-S value of 1 (see methods and supplementary notes for behavioural validation of this approximate threshold, Supplementary Figs. 11–14). For Phosphor-Converted (PC) amber LEDs, their impact varies with decreasing light levels, yielding similar or even higher chromatic contrasts than moonlight at high light levels, yet performing much worse than moonlight at lower light levels. High-pressure sodium and metal halide lights follow this pattern to a lesser extent: they perform similarly to moonlight when hawkmoths are viewing flowers in the white/yellow group, but when pink/purple flowers are considered they switch from enhancing chromatic contrasts at the highest light level to reducing them at lower light levels.

**Intra-specific communication**. The functions of colourful wing markings on hawkmoths are poorly understood, but may be used for intraspecific species identification and sexual signalling. We modelled the effect artificial light might have on intraspecific signalling in 14 species of crepuscular and nocturnal European hawkmoths (Supplementary Fig. 1, Supplementary Table 2) by measuring the reflectance spectra of the wings of museum specimens (Supplementary Fig. 2) and using the above modelling procedures. As with flower detection, the specific effect of light type on maximum chromatic contrasts in each specimen's wings varied with the colours viewed, with a significant interaction between species and light type at every light level ($\chi^2 = 748.04$, $\chi^2 = 885.28$, $\chi^2 = 947.47$, $\chi^2 = 1066.6$, $\chi^2 = 1136.5$ respectively for light levels from 10 to 0.001 cd m$^{-2}$, df = 104, $p < 0.001$). Overall, broadband white artificial lights perform either similarly to or better than moonlight at all light levels tested (Fig. 1b; Supplementary Fig. 3a). Again, the narrow-band orange LED and low-pressure sodium lights prevent discrimination of even the greatest

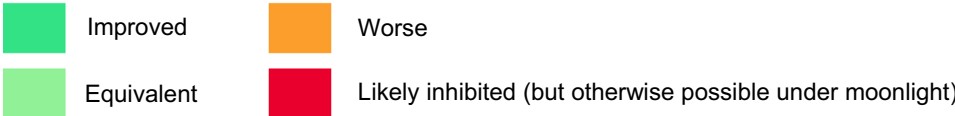

**Fig. 1 Effects of artificial lighting on colour perception, compared to moonlight at equivalent light levels.** Lights are labelled as performing better (green) or worse (orange/red) than full moon conditions, from the hawkmoth perspective. Hawkmoth perception was modelled for: **a** contrast of white/yellow flowers (top of cells), and pink/purple flowers (bottom of cells) against natural foliage backgrounds; and **b** maximum internal contrast for intraspecific signals (fore- and hindwings, 14 species positioned alphabetically in each cell). Blue tit perception was modelled for: **c** hawkmoth internal contrast (signalling, fore- and hindwings); **d** hawkmoth forewing camouflage against green (top of cells) or brown (bottom of cells) natural backgrounds; and **e** hawkmoth background selection under artificial light followed by blue tit detection under diurnal predatory search behaviour. Thresholds for (**a**) and (**d**) are set where there is no overlap between confidence intervals compared to full moon conditions. Thresholds in (**b**) and (**c**) are determined by the combined estimated effect of light type and the light:species interaction in models of the impact of lighting on maximum chromatic contrasts, with thresholds for difference from moonlight met when the absolute value of this estimate is greater than 1 Delta-S. Thresholds for (**e**) are based on statistical differences against full moon conditions representing colour mismatches >1 Delta-S. For white and PC amber LEDs, the statistical results in **b**, **c** and **e** are based on an LED in the middle of our CCT range (CCT = 3079 K) and PC amber Cree respectively, so conclusions may not apply to white LEDs with extreme CCT values.

chromatic contrasts in the wing patterns, consistently yielding Delta-S values below 1. Following the trend for flowers and vegetation, chromatic contrasts are enhanced under high levels of illumination by PC amber LEDs compared to moonlight, but the effect is reversed at light levels of 0.01 cd m$^{-2}$ and below (Fig. 1b). These results are also broadly supported by analyses of areas occupied by individuals' colours in the hawkmoth RNL chromaticity space (Supplementary Fig. 4).

**Anti-predator behaviours**. Wing colours may also be involved in mitigating predation risk, by way of either concealment or

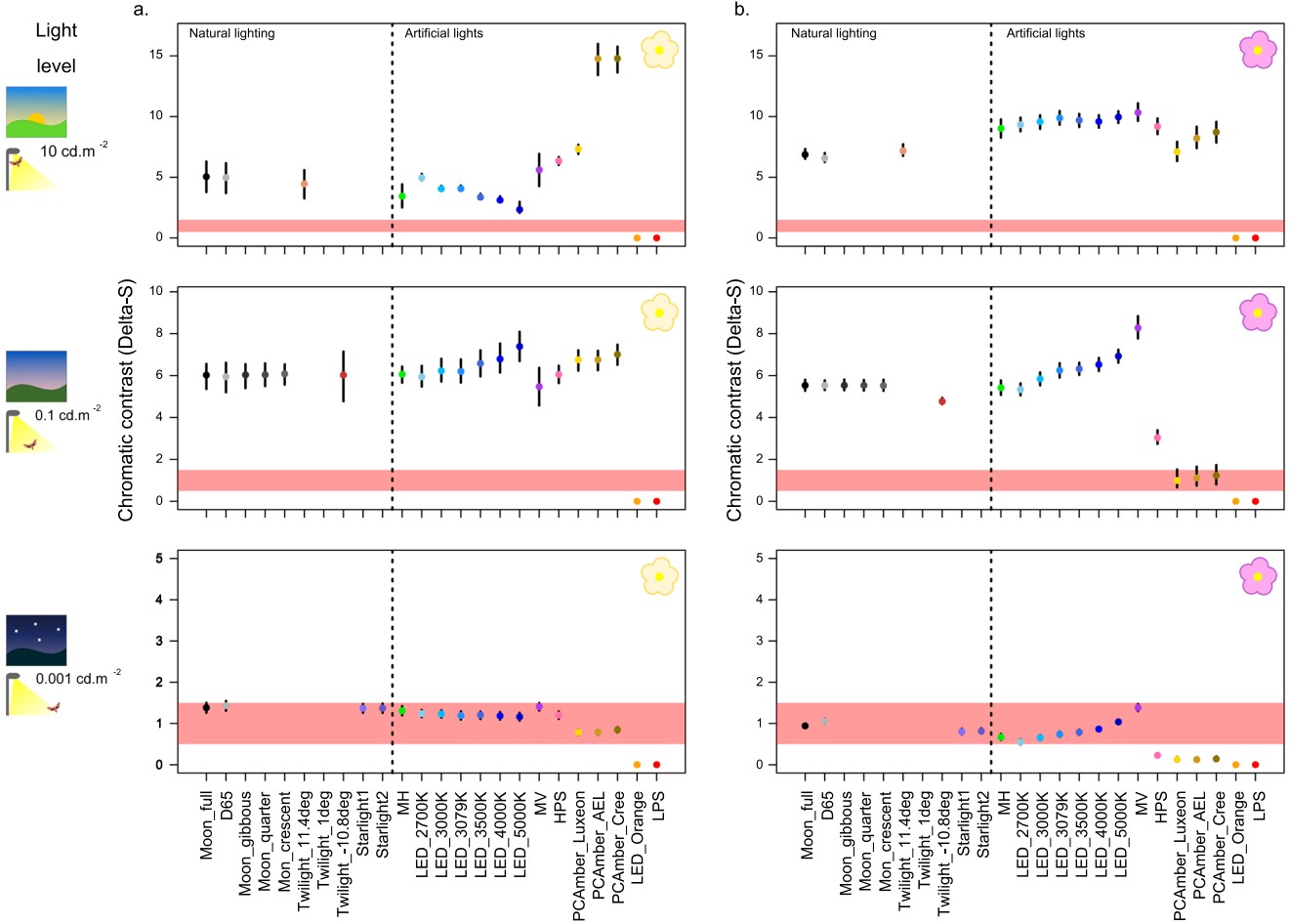

**Fig. 2 Flower colour discrimination by moths under artificial light.** Bootstrapped estimates of chromatic distance (in Delta-S) between geometric means of floral colours ($N_{white/yellow} = 60$, $N_{pink/purple} = 80$) and background colours ($N_{backgrounds} = 953$) in the hawkmoth RNL chromaticity space. Mean and 95% confidence intervals are shown for white/yellow flowers (**a**) and pink/purple flowers (**b**), under three light levels (10, 0.1 & 0.001 cd m$^{-2}$, from top to bottom, represented by cliparts on the left-hand side), based on 1000 bootstrapped replicates per comparison. Each light type is represented by a different colour. The coloured bands represent putative thresholds for colour discrimination (Delta-S between 0.5 & 1.5). Full moon and D65 conditions are considered for every light level to provide consistent references, and additional relevant natural light types are included for each light level as appropriate. Source data are provided as a Source Data file.

communication with predators. Camouflage in particular is the most widespread form of anti-predator defence, playing a key role in determining animal survival[16], and evolution[17,28], and is the most likely function of wing coloration in prey animals such as hawkmoths. Birds are considered to be potential predators of hawkmoths[29] and several avian species extend their activity at dawn and dusk around artificial lights[30], including for foraging[31]. We therefore modelled how artificial lights might impact the perception of hawkmoth wing coloration by a typically diurnal passerine (the blue tit *Cyanistes caeruleus* UVS visual system[32]), under low light conditions (photon shot noise[24]; Fig. 3; Supplementary Fig. 7). When viewing the generally dull forewings of hawkmoths at rest on natural backgrounds, avian perception of colour contrasts between moth colours and both green and brown backgrounds would be restricted to the highest level of illumination modelled (assuming the chromatic discrimination threshold is near 1 Delta-S, see methods and supplementary notes for behavioural validation of this assumption Supplementary Fig. 14). Moreover, no artificial lights enhanced the ability of birds to detect chromatic contrasts between the moth forewings and the backgrounds, compared to an equivalent level of full moonlight; most light types performed similarly to moonlight. Moreover,

mercury vapour, low-pressure sodium, and orange LED lights actually reduced perceptible contrasts (Figs. 1d and 3), so the ability of moths to hide from avian predators at night is unlikely to be compromised by artificial lighting.

Alternatively, hawkmoth wing patterns can also include conspicuous colours, particularly on the hindwings, and these may be used for signalling to predators. Although some of their caterpillars feed on chemically defended host plants, there is no evidence that they sequester any toxins or use them for their own defence as adults[33–35], so any signalling role for wing colours is more likely to function through Batesian mimicry or as a startle display (as in the eyed hawkmoth *Smerinthus ocellata*[36]) rather than aposematism[37]. Based on the same model passerine system, perception of chromatic contrasts between any colour patches on hawkmoth wings (including on the more colourful hindwings) by diurnal birds would break down below levels of illumination equivalent to 1 cd.m$^{-2}$ (Fig. 1c; Supplementary Fig. 3b). There was a significant interaction between the species viewed and light type at all light levels tested ($\chi^2 = 721.22$, $\chi^2 = 726.76$, $\chi^2 = 728.07$ respectively for light levels from 10 to 0.1 cd m$^{-2}$, df = 104, $p < 0.001$) but, in general, at equivalent light levels, no type of artificial lighting substantially changed avian perception of colour contrasts in the wing patterns compared to moonlight (Fig. 1c;

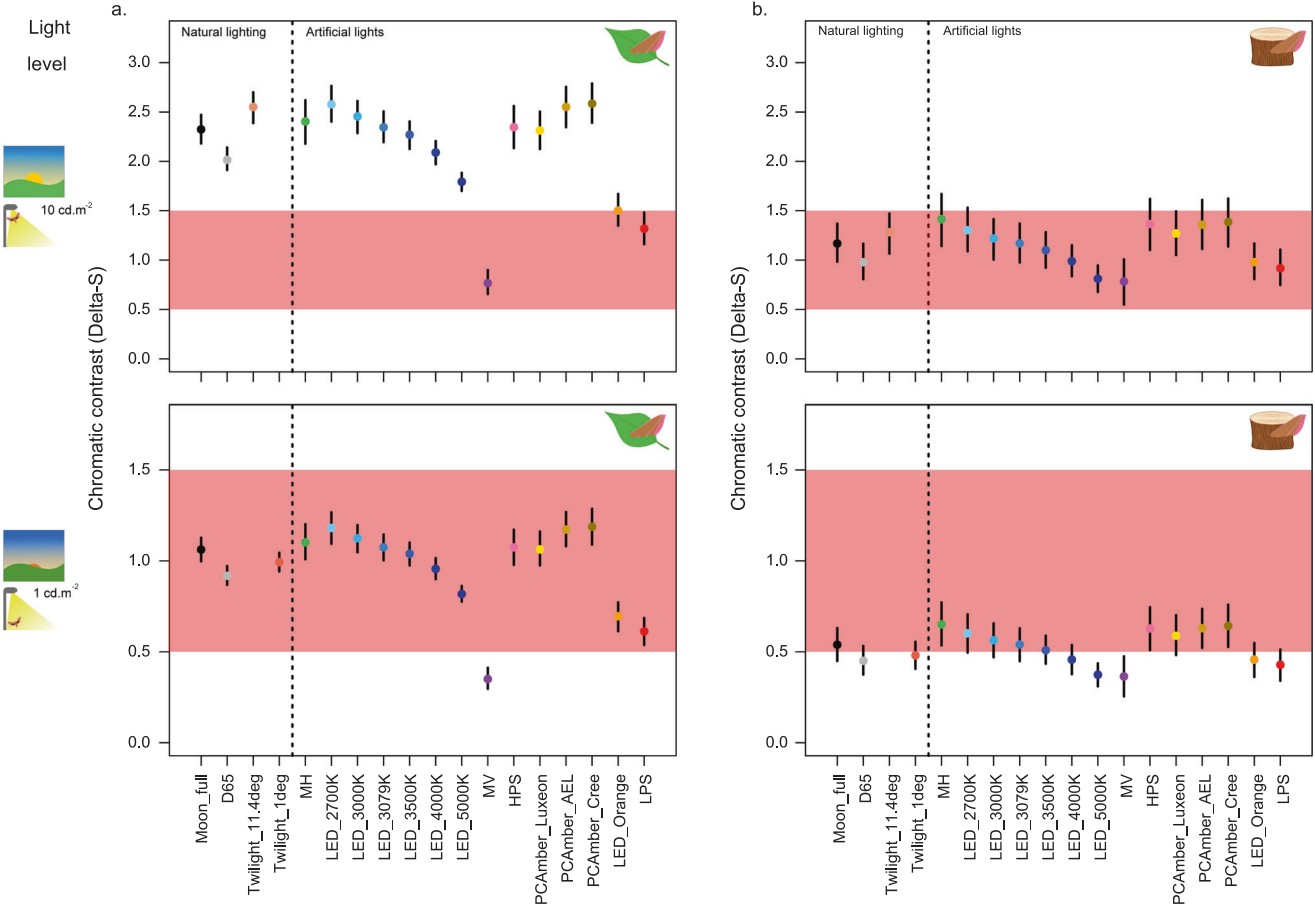

**Fig. 3 Moth-background colour discrimination for birds under artificial light.** Bootstrapped estimates of the distance (in Delta-S) between the geometric means of moth forewing colours ($N_{moths} = 235$) and background colours ($N_{green} = 871$, ($N_{brown} = 274$), in the blue tit RNL chromaticity space. Mean and 95% confidence intervals are shown for green (**a**) and brown (**b**) natural backgrounds, under two light levels (10 and 1 cd m$^{-2}$, from top to bottom, represented by cliparts on the left-hand side), based on 1000 bootstrapped replicates per comparison. Each light type is represented by a different colour. The coloured bands represent putative thresholds for colour discrimination (Delta-S between 0.5 and 1.5). Source data are provided as a Source Data file.

Supplementary Fig. 3b). The few exceptions to this rule are enhanced contrasts for some species under metal halide, mercury vapour, high-pressure sodium and PC amber LEDs, and reduced contrasts for *D. elpenor* wing patterns under low-pressure sodium lights, at the highest level of illumination. As for estimates of volumes occupied by each moth's wing colours in the tetrachromatic avian RNL chromaticity space, all artificial lights yield smaller volumes than under moonlight, with particularly large effect sizes for orange LED and low-pressure sodium lamps (Supplementary Fig. 5). Therefore, with the exception of some specific moth colours, artificial lighting does not facilitate detection by avian predators of camouflaged hawkmoths, nor does it substantially enhance the moths' potential visual signals.

While we find little impact of artificial lighting on avian predation risk at dawn and dusk, altering the moths' visual environment at night may have more significant consequences on their exposure to predators during the day. Recent research has demonstrated that a number of animals improve their concealment by choosing backgrounds which complement their appearance[38], and moths are thought to use vision, among other cues, to select optimal colour-matching resting locations[39,40]. Artificial light has the potential to create dramatic colour mismatches between moths and their backgrounds, because the moths must select resting locations under the influence of artificial light (before dawn), while the majority of visually guided predators will then be active under daylight. We modelled this

effect by comparing the moth forewing colours to a database of natural reflectance spectra (including bark and leaves), as viewed by hawkmoths under each light source at a range of intensities. All pairs of moth and background colours found to be an excellent colour match for hawkmoth vision (i.e. indiscriminable, or below a putative threshold of 1 Delta-S) were then modelled using the blue tit visual model under daylight (D65) conditions. For all levels of nighttime lighting considered, the type of lighting under which pairs of colours were assessed as perfectly matched to the background for hawkmoth vision had a significant effect on levels of contrast perceived by avian predators in daylight ($\chi^2 = 35560$, $\chi^2 = 30285$, $\chi^2 = 23788$, $\chi^2 = 7337.6$, $\chi^2 = 1337.4$ respectively for light levels from 10 to 0.001 cd m$^{-2}$, df = 8, $p < 0.001$). With the exception of metal halide and mercury vapour lights, selecting colour matches to the hawkmoth visual model under artificial lights, as opposed to moonlight, consistently led to significantly poorer colour matches between moth colours and natural backgrounds to avian vision in daylight, though effect sizes were generally very small, especially at lower light levels (Fig. 1e; Supplementary Fig. 8). Monochromatic lights had the strongest effect, causing an increase in colour mismatch to diurnal birds greater than 1ΔS compared to moonlight (Fig. 1e), suggesting that choosing resting backgrounds under relatively high levels of some artificial lights could result in inappropriate choices and an increased risk of detection by diurnal predators (Supplementary Fig. 8).

## Discussion

This study highlights the complex ways in which artificial illumination can affect visual ecology. By comprehensively modelling the effects of light emission spectra, light intensity, surface reflectance and receiver vision we have revealed a range of previously unforeseen relationships. The most striking finding is that broadband amber light sources (such as PC amber LEDs and high pressure sodium) are predicted to have a disruptive effect on hawkmoth flower colour perception. At high intensities these light sources can provide very good colour discrimination for finding flowers, however, under lower intensities (typically found tens of meters from a light source, or under skyglow) the same light source can actually inhibit colour discrimination. This is likely to interfere with a hawkmoth's ability to remember and efficiently handle flowers. Narrow-band sources (such as LPS and orange LED) prevent perception of colour contrasts in moth wings and between flowers and vegetation, and may result in poor background resting location choices, leaving the moths more vulnerable to diurnal predators.

A variety of approaches to reducing the environmental impact of artificial nighttime lighting have been proposed[41,42]. In the main these concern quite generic changes to reduce the spatial and temporal occurrence of such lighting, limit its intensity and to limit the use of broader spectrum and blue-rich lighting[20]. Animal visual ecology is an increasingly important factor in these recommendations, as recent evidence from comparisons between emission spectra of artificial lights and behavioural responses or spectral sensitivities of different species suggest that broad-spectrum lights are most likely to disrupt ecological interactions[18,20,21]. By contrast, amber LEDs have been seen as less harmful[20], and are being deployed with, still low but, increasing frequency. This approach is based on a range of mechanisms, notably observations of insect phototaxis that show high capture rates in light traps with high blue/UV output[43–45], and fitted models that suggest that the UV and blue photoreceptors in insects are key in driving phototaxis[46], as well as the effects of blue-rich light in suppressing melatonin production across a wide range of taxa[47]. However, artificial nighttime lighting can interfere with insect ecology in a wide variety of ways[8], and potential solutions for one problem may be inappropriate for others; for example, narrow-band long-wavelength lighting may reduce interference with bioluminescent signals, such as those of some fireflies[13], but inhibit perception of colour signals by other insects. More comprehensive assessments of the effects of spectral composition of light sources on visual ecology, and hence on aspects of behaviour such as foraging, predation rates, and mate selection have not previously been undertaken. The results reported here, suggesting previously under-appreciated effects of amber lighting on the visual ecology of valuable nocturnal pollinators, argue for more in-depth assessments of the impacts of specific lights on relevant ecological interactions, and a more nuanced approach to solutions for mitigation.

## Methods

**Spectral measurement**. To quantify colours, as viewed by either hawkmoths or avian predators, under different light types, we combined reflectance measurements with emission spectra from several different kinds of light sources to carry out low-light visual modelling specific to each visual system. Reflectance measurements from hawkmoth wings and natural backgrounds were taken with an Ocean Optics USB2000 + spectrometer (Dunedin, FL, USA), with the probe held at a 45° angle, coupled to either a PX-2 pulsed xenon lamp or a stabilized deuterium light source (SLS204, ThorLabs, Newton, NJ, USA). Five specimens each of 14 nocturnal UK species of hawkmoths (Lepidoptera: Sphingidae) were measured in 2018 ($N = 70$), using specimens in the collections of Bristol Museum & Art Gallery, Exeter Royal Albert Memorial Museum (UK) and private collections (see Supplementary Table 2 for details of specimen provenance). Based on visual inspection of their wing patterns, 4–9 distinct colour patches were selected per species

(in total, $N_{MOTH} = 445$, Supplementary Figs. 1 and 2). Samples from 14 plant species with flowers visited by adult hawkmoths were collected in and around Penryn, Cornwall (UK) in 2018 and 2019 (Supplementary Table 3); 10 independent samples of the main floral and leaf colours were measured from each species ($N_{FLOWER} = N_{LEAF} = 140$ each for flowers and leaves). Leaf measurements were combined with data from the Floral Reflectance Database (FReD[48]; all search results for "Leaf", $N_{FReD} = 513$), measurements of samples from 9 species of grasses (collected in Penryn, Cornwall, including leaf blades and seed heads both fresh and dry, $N_{GRASS} = 300$) and natural background spectra from the MICA image analysis toolbox[49] ($N_{BGD} = 192$) to create a comprehensive dataset of green and brown natural background spectra ($N_{NATBGD} = 1145$).

We modelled the effects of 10 natural illuminants, including natural moonlight, twilight, starlight and daylight spectra, and 14 artificial light sources, including several traditional artificial lighting types, white LEDs in a range of correlated colour temperatures (CCTs) and phosphor-coated (PC) amber LEDs (see Supplementary Table 1 for data collection and light spectra). To reduce the effect of noisy measurements, emission spectra were thresholded so values below 1/100th of the maximum radiance per light type were set to a very low but non-zero value ($10^{-16}$). We used known spectral sensitivities for photoreceptors of the trichromatic elephant hawkmoth *Deilephila elpenor*[14] and the tetrachromatic UV-sensitive model passerine, the blue tit *Cyanistes caeruleus*[32], to model colour perception by hawkmoths and potential avian predators respectively.

**Analyses**. All visual modelling and statistical analyses were carried out in R v3.5.2[50] and a full script is provided in the Supplementary Information. In brief, we calculated absolute quantum cone catch values for every sample patch (moth wings, flowers and natural backgrounds) under every light type, for every hawkmoth[14] and blue tit[32] photoreceptor type. Modelling the photon catch of hawkmoths or birds requires information on the quantal flux of photons reaching the animal's photoreceptors. This calculation combines information on the photonic emissions of the light source at each wavelength, the number of photons being reflected from the surface, and the number of photons transmitting through the optical media and then being absorbed by the photopigments (see supplementary methods for equations and parameters). We repeated these calculations for five human-based illumination intensities (from 10 to 0.001 cd m$^{-2}$), corresponding to light levels from dusk to starlight, the lower limit of colour vision for *D. elpenor*[23]. Typical artificial streetlight intensities also cover this range, dependent on distance to the light. Only the three highest illumination levels were applied to the blue tit visual system, in accordance with their estimated threshold for colour vision in dim light (10 to 0.1 cd m$^{-2}$[51]). Based on the quantum cone catches, we then calculated coordinates in the hawkmoth and blue tit receptor noise-limited (RNL) spaces under low light conditions, and chromatic contrast based on Delta-S between pairs of colours as appropriate[24,27]. The RNL model is most appropriate for chromatic differences near the threshold point, however, our modelling often results in suprathreshold values. Nevertheless, any two adjacent suprathreshold colours must blend to become sub-threshold at a viewing distance dependent on the receiver's acuity limits. Therefore, while our modelling may not be ideal at close range, the suprathreshold values will scale with critical maximum detection distance, which will be larger with greater colour contrasts (we illustrate this spatiochromatic effect in Supplementary Fig. 10). To assess how artificial lighting affects the perception of chromatic contrasts between colours in hawkmoth wing patterns, we calculated the maximum chromatic contrast between colours, and the area or volume occupied by all colours in the RNL space, per moth specimen, light type and light level, as perceived by the hawkmoth and blue tit visual systems respectively. For the purpose of statistical tests, white and PC amber LEDs were represented by a single light type each (white LED with CCT = 3079 K and PC amber Cree, respectively). Effects of light type on maximum contrasts were tested with a simple mixed model per light level, allowing an interaction between light type and moth species, with specimen as a random effect, using the package lme4[52].

To compare the visibility of flowers against vegetation under different lights, we calculated bootstrapped estimates of the distance between the geometric means of flower and background colours (all leaves and grasses, $N_{LEAF} + N_{BGD} + N_{FReD} = 953$ samples) in the hawkmoth RNL space for each light type and light level (see Supplementary Fig. 6), following methods in Maia & White[53], modified to use our scotopic Delta-S calculations, using the bootcoldist function in the package pavo;[54] our modified bootstrapping function is included in the supplementary R analysis script. In essence, this method repeatedly samples the colour differences between one flower spectrum and one background spectrum, calculating the bootstrapped confidence intervals for mean distances between the two sets. After visual inspection of the data, floral colours were split into two broad colour groups for the analysis (pink/purple and yellow/white, $N_{PINK} = 80$ and $N_{WHITE} = 60$ respectively). Evening primrose (*Oenothera* sp.) samples do not group so closely to other white/yellow flowers, but were included as results did not vary qualitatively with or without these species. The same method was used to quantify the visibility of moths against natural backgrounds (separated into green and brown backgrounds, $N_{GREEN} = 871$, $N_{BROWN} = 274$) as perceived by the blue tit visual system; here, only forewing colours ($N_{FOREWING} = 235$) were included in this analysis, as hawkmoth hindwings are generally hidden at rest (see Supplementary Fig. 7).

Finally, moths must choose where to rest at night (potentially under the influence of artificial light) but rely on a day-time colour match to their

background to protect them from predators. To quantify this effect we calculated chromatic contrasts between every moth forewing colour ($N_{FOREWING} = 235$) and natural background colour ($N_{BGD} = 953$) for hawkmoth vision in low light as described above, under each combination of light type and light level. For each light level and light type combination, we then selected only pairs of colours that would be indiscriminable to hawkmoth vision (Delta-S < 1) for analysis (simulating an excellent background choice to hawkmoths under those viewing conditions). To model equivalent contrasts for blue tit vision in D65 conditions, cone catches to blue tit photoreceptors were calculated with the vismodel function and built-in blue tit data in the pavo package[54], with an ideal transmission medium and the von Kries transform, then chromatic contrasts in the RNL space were obtained using the Vorobyev-Osorio receptor noise-limited model, with a Weber fraction $\omega = 0.05$[22] for the most abundant receptor class. The corresponding chromatic contrasts for blue tit vision in D65 conditions were analysed using linear mixed effects models with light type as a fixed effect and moth and background colour IDs as random effects, and planned comparisons to moonlight were implemented as described previously (Supplementary Fig. 8).

**Reporting summary**. Further information on research design is available in the Nature Research Reporting Summary linked to this article.

## Data availability
Source data are provided with this paper.

## Code availability
Full analysis code is available as a supplementary information file.

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

## Acknowledgements

We thank R. Rowson at Bristol Museum and H. Morgenroth at Exeter RAMM for access to museum collections, L. Kelley and A. Spalding for the loan of their specimens, A. Kelber and J.P. Renoult for assistance with model equations, S. Johnsen for making the elephant hawkmoth photoreceptor sensitivity curves and natural lighting spectra available, and J. Chapman for evening primrose samples. Work was supported by Natural Environment Research Council grants NE/P018084/1 (to J.T.) and NE/P01156X/1 (to K.J.G. and J.B.).

## Author contributions

K.J.G. and J.T. conceived the study. E.S.B., K.J.G. and J.T. designed the analysis methods. E.S.B., E.R. and J.B. collected and analysed the data. E.S.B. prepared the figures. E.S.B. and J.T. wrote the methods, and the first draft with K.J.G. All authors contributed to further drafts.

## Competing interests

The authors declare no competing interests.
