## [Peer Review File · Nature Communications]

REVIEWER COMMENTS

Reviewer #1 (Remarks to the Author):

The authors present a very comprehensive study on the effects of different anthropogenic light on biologically meaningful visual stimuli (e.g. substrate and conspecifics) in the eyes of a hawk moth as well as how an avian predator would perceive this potential prey item - all compared to moonlighting at different light levels! Pretty awesome approach and with a very important organism - the hawk moth, which is an important nocturnal pollinator and currently known to have the most sensitive color vision. I do commend the authors on their large data set and comprehensive approach. Furthermore the manuscript is well written and flows well. The figures are all excellent and informative. I especially like the first figure. I also like supplemental figure 3 and if possible, I think that figure should be included in the main text.

This study is an excellent example on how to incorporate visual ecology into the large discipline of ALAN (artificial light at night) and will inspire many "copycats". Because this is such an excellent approach and will set the standard for understanding the effects of light pollution on visual ecology, I do have one major concern. All of the contrasts and findings rely upon comparing to moonlight conditions and yet I couldn't find any information on their moonlight measurements other than a quick note in Table S1 which shows that they measured moonlight using a Specbos 1211 spectroradiometer and measured radiance of the moon. This is a huge red flag, but not a kiss of death. The authors need to address these issues before this manuscript is ready for publication in Nature or any other high impact journal.

1) You need to have a section on what moonlight means. What phase/percent illumination? How high in the sky was it? These matter a lot for overall spectra, so it needs to be noted.

2) You need to caveat that you measured radiance instead of irradiance, which of course you would need to do because if you measured irradiance you would mostly get ALAN and not moonlight - please make this clear to the reader as I suspect most readers won't know this.

3) Depending on the answer in number 1, you need to discuss what your assumption of using moonlight at natural nighttime lighting means for your paper. Unfortunately, I think that since you used moonlight as your only night time light reference, you are missing a big picture of the natural conditions for these moths. We know that starlight is in fact much more peaky and results in an irradiance closer to that of HPS - it would be very informative to run your models with starlight spectrum (see Optics of Life for that spectrum) and see how these ALAN spectra compare. I realize that this will be a lot of work, but you would nail this study out of the park and cover all of the bases! If you decide that including the starlight spectrum is too much work, you must discuss that you used moonlight (which represents less than 50% of nighttime lighting as the moon is below the horizon for a lot of nighttime) and thus you are only able to make comparisons of natural conditions for when there is moonlight (again, less than 50% of night). I do hope you will do the model with starlight (in which case send it to Nature or Science and ask me to be a reviewer...) as I think it will show that those red shifted anthropogenic lights are similar to starlight.

Small things:

1) I don't understand what you are trying to communicate in 294-296. What do you mean you only

selected pairs of colours with a perfect colour match - huh? I don't understand and I know I am not giving you much to work with, wish we could talk on the phone...

Figure S2 - what are the patches? You have numbers but I didn't see a legend for what the numbers mean. I also understand that the different species of moths have different numbers but perhaps you could say that the numbers start at one wing and then go clockwise or something like that?

That's all. I really do think your paper is excellent and absolutely needed, I just wish starlight was in there or you discuss why you only need to think about moonlight with regards to the hawkmoth, which I believe is not the case.

I hope to see a revision,
Brett Seymoure

Reviewer #2 (Remarks to the Author):

This paper addresses the effects of various forms of artificial lighting on color perception in a moth and a bird under various intensities and contexts. It is well written, and the methods (to the extent that they can be determined -- see below) are solid. The results themselves are relatively unsurprising, that more monochromatic lights have more effect of color perception and that this effect depends on the lights' spectra and intensities, but it is good to see a systematic and comprehensive approach to this problem. I have a few concerns:

1. the most major perhaps is that many of the color distances are so far above threshold that moderate to minor decreases in them should make little difference. I point the authors to this year's ProcB paper by Santiago et al that confirms what has been long assumed -- that color distances well above threshold aren't really meaningful except in certain specific contexts (e.g. attenuating media). Detection, as is typical, follows a sigmoidal curve that asymptotes once well above threshold. So I'm unsure which of the "improved" or "worse" conditions actually have meaningful effects, especially since this paper is entirely modeling without behavioral data. I am willing to believe the lighting conditions that completely destroy color perception (e.g. LPS), but this of course has been well known. There are a very large number of papers now that take a set of reflectances and illumination spectra and then calculate delta-s using the RNL model, but very few that actually groundtruth this work. Ideally, the authors should address this with behavioral trials, but at a minimum they need to find some way to tie these differences to reality, and to think carefully about which differences may truly matter.

2. The methods section is relatively opaque, since it mostly refers to data being pipelined through a number of software packages such as Pavo. This is of course done in some other fields -- bioinformatics being a major offender -- but the visual ecology world is small, so it is highly unlikely that these software packages will still exist in ten year's time. Therefore it would be good to show the actual math of what is being done, since the equations will be understandable hundreds of years from now, if the last few

hundred are any indication. It's also worth noting that not every researcher has access to the materials needed to run these packages, even if the packages themselves are freely available. Also, most creators of software packages provide no system that guarantees that their programs will still be available and supported in the future. The basic algorithms and math are fairly universal, but the syntax can fail when the underlying R code (for example) goes through a major update, even assuming that the program itself is safely archived. These are complex issues, but it's worth an author's time to present their methods in a form that will be usable in the future.

Reviewer #3 (Remarks to the Author):

This is a beautifully written and presented paper investigating how artificial lights may alter the perception of hawkmoth colour signals by conspecifics and potential predators. With visual modelling of spectral reflectance measurements of hawkmoths and potential backgrounds the hawkmoths could be viewed against, and the illumination spectra of artificial lights, the authors present theoretical evidence of how colour signals may be perceived differently.

I have no issues with the way that the data was collected, modelled and analysed. These are very clear and presented well. However, I do have major reservations about the relevance and interpretation of chromatic contrast values calculated in the study without further behavioural validation based on reasons below.

The RNL model is commonly used to quantify chromatic contrast of colours viewed by a given visual system under specific conditions, and under these conditions the discrimination threshold is assumed to be $\Delta S = 1$. However, the relationship between distance in colour space and perceived colour difference is not straightforward, and there has been little behavioural calibration of the model with broadband colours, such as the ones presented in this study. Recent studies that have attempted this in different types of fish have shown deviance from RNL model predictions, and that discrimination thresholds can vary between 0.7- > 3 depending on directions in colour space, see:

- o Sibeaux, A., Cole, G. L. and Endler, J. A. (2019). Success of the receptor noise model in predicting colour discrimination in guppies depends upon the colours tested. *Vision Research* 159, 86-95.
- o Escobar-Camacho, D., Taylor, M. A., Cheney, K. L., Green, N. F., Marshall, N. J. and Carleton, K. L. (2019). Color discrimination thresholds in a cichlid fish: *Metriaclima benetos*. *Journal of Experimental Biology* 222, jeb201160.
- o Cheney, K. L., Green, N. F., Vibert, A. P., Vorobyev, M., Marshall, N. J., Osorio, D. C. and Endler, J. A. (2019). An Ishihara-style test of animal colour vision. *Journal of Experimental Biology* 222, jeb189787. Discrimination thresholds are even higher when moving away from the neutral (achromatic) point for more highly saturated colours (which is also the case with humans).

Therefore, while it is clear that hawkmoth can perceive colour in dim light conditions, calibration of the model with behavioural experiments to show discrimination threshold are ~ 1 , has not been done (to my knowledge). It is therefore hard to interpret the measurements being obtained here, which often fall in

the range of 0-2.5 (ΔS)(as per Figure 3).

Furthermore, 'suprathreshold' colours also have to be interpreted with caution once they go beyond the discrimination threshold. This may impact interpretation of whether colour discrimination is 'improved' or 'worse' (as per Figure 1). See recent paper: Santiago et al. (2020). Does conspicuousness scale linearly with colour distance? A test using reef fish. *Proceedings of the Royal Society B: Biological Sciences*, 287 (1935) 20201456, which addresses this issue.

Regards, Karen Cheney

Reviewer #4 (Remarks to the Author):

This paper investigates the effect of artificial light at night on colour discrimination by hawk moths. The authors address this topic thoroughly by modelling chromatic contrast under a variety of different lighting scenarios and the results suggest that some types of artificial light will hinder colour discrimination and potentially a moth's ability to background match. The paper is written very nicely and the figures are fantastic. Artificial light at night is a very important topic and any effects on insects may help us to understand declines in insect numbers. Therefore, this topic is likely to be of interest to a broad range of scientists.

Before recommending this study for publication, I have some questions and comments that I think should be addressed. Most of my questions are regarding how well the models mimic environmental conditions. I am not sure that all the intensities modelled are relevant to actual conditions and some of the comparisons may not directly address the research question (detailed below). More information regarding the biology of the system would also be useful. Some of these changes may change the results and interpretation.

Comments on illumination choices:

I am interested in the choice of intensities of the lights for a few reasons. Firstly, why was moon light and D65 modelled at several different intensities? These light sources do not vary by multiple orders of magnitude in nature, so the variation isn't ecologically relevant. It would be more beneficial to keep these at natural intensity levels so the reader can see what chromatic contrast would be under natural conditions without any interference of artificial light. This would better address the research question of how artificial lights affect chromatic contrast compared to natural conditions.

Following this, it might be more useful to model dusk irradiance rather than D65 since the highest luminance of 10 cd m² is similar to dusk light levels. I would also be interested to see starlight irradiance since you mention that these moths can discriminate colours under starlight conditions.

It would also be great to have some more information on the intensity of different artificial light sources. I imagine they are not all the same intensity in the field – do they vary greatly? Furthermore, the lowest luminance measurement corresponds to low level skyglow. Therefore, it is relevant to model that for specific light sources when skyglow is likely a mix of multiple light sources? It may be more relevant to obtain a measure of skyglow (potentially with and without moonlight) for these models. Would moonlight mix with artificial light sources at other intensities also?

Other comments:

I am also curious as to why the von Kries transformation was not applied. A previous study investigated contrast for these moths with and without the von Kries transformation (Johnsen et al 2006), and Kelber et al 2003 behaviourally demonstrated colour constancy. This would likely have a large effect on the results. Sorry if I have misunderstood!

More natural history about the system would also be useful to understand the conditions selected in the model. For example, are they often found near the lights or right under the lights? Are the flowers often found near/under lights? Do they fly at dusk or only night time?

It would be great if you could add in a paragraph discussing achromatic discrimination. I am curious whether the moths could compensate for any loss in colour discrimination via achromatic discrimination.

I like the logic behind your models of background matching! However, I'm unsure about the interpretation. It looks like the moths don't match the background, even if they select their background under moonlight conditions. Chromatic contrast may not be linear at these high JNDs, so the increase may not be meaningful (c.f. Santiago et al 2020, Proc R Soc B). So the artificial lights might not make it much worse.

Minor comments:

Is this study relevant to other nocturnal/crepuscular organisms? Are there others that use colour vision?

Are each of the light sources chosen commonly found in the human modified environments?

Line Comments:

Line 30: Any references for sexual signalling? Maybe fireflies?

Figure 1: to take up less space you could split the legend into two columns to fill up the white space to the right. Also, it looks like only 7 species in each cell for b&c? Really great figure!

Line 63: Was the background foliage a combination of many plants?

Line 217-220: Are the colours structural? If they are pigment based, do they match alive hawk moths? Some are quite old and the pigments might have started to break down.

Line 250: What intensities are relevant to artificial lights? I see it is mentioned in the supp info (Lines

396-298. I think this needs to be included in the main text to provide context.

Line 284-299: It would be good if you could briefly restate the logic here (i.e. a moth chooses where to sit at night, but needs to match throughout the day for diurnal predators).

I hope you find this helpful.

The four reviews we received were universally positive, constructive and thorough, and we would like to thank the reviewers for their excellent suggestions. The reviewers were also broadly in agreement over the weaknesses of our previous manuscript, and as such we have implemented some major changes, involving adding new light sources and a complete reanalysis of all of our data. We have also performed some additional modelling to offer some behavioural validation to our methods, and spatiochromatic modelling to highlight the issue of suprastimulus colours. This major undertaking has resulted in a much stronger manuscript. Below we respond to each reviewer's points in *blue italics*.

REVIEWER COMMENTS

Reviewer #1 (Remarks to the Author):

The authors present a very comprehensive study on the effects of different anthropogenic light on biologically meaningful visual stimuli (e.g. substrate and conspecifics) in the eyes of a hawk moth as well as how an avian predator would perceive this potential prey item - all compared to moonlighting at different light levels! Pretty awesome approach and with a very important organism - the hawk moth, which is an important nocturnal pollinator and currently known to have the most sensitive color vision. I do commend the authors on their large data set and comprehensive approach. Furthermore the manuscript is well written and flows well.

We are delighted to have received such a positive appraisal.

The figures are all excellent and informative. I especially like the first figure. I also like supplemental figure 3 and if possible, I think that figure should be included in the main text.

Many thanks – we are glad the figures are informative. On balance we believe Figure S3 is probably best left in the supplementary section given competition for space in the main manuscript. This figure highlights the ability of hawkmoths and birds to see the internal colours within just one example species (the elephant hawkmoth). While this is an important aspect of visual ecology, we feel that the pollination data (Fig 2) and camouflage data (Fig 3) are likely to be more important for hawkmoth survival, and are able to show the effects across all species.

This study is an excellent example on how to incorporate visual ecology into the large discipline of ALAN (artificial light at night) and will inspire many "copycats". Because this is such an excellent approach and will set the standard for understanding the effects of light pollution on visual ecology, I do have one major concern. All of the contrasts and findings rely upon comparing to moonlight conditions and yet I couldn't find any information on their moonlight measurements other than a quick note in Table S1 which shows that they measured moonlight using a Specbos 1211 spectroradiometer and measured radiance of the moon. This is a huge red flag, but not a kiss of death. The authors need to address these issues before this manuscript is ready for publication in Nature or any other high impact journal.

We have taken this important concern on board, and as such have re-run the entire data analysis with a large range of natural illuminant spectra (more below).

1) You need to have a section on what moonlight means. What phase/percent illumination? How high in the sky was it? These matter a lot for overall spectra, so it needs to be noted.

2) You need to caveat that you measured radiance instead of irradiance, which of course you would need to do because if you measured irradiance you would mostly get ALAN and not moonlight - please make this clear to the reader as I suspect most readers won't know this.

Both these concerns are dealt with by using moonlight measurements from Sönke Johnsen, previously used in a study of hawkmoth visual perception (Johnsen et al. 2006). Instead of a single moonlight measurement, we now use four moonlight irradiance spectra, corresponding to a full moon (elevation 69°, moon 98% full; see Johnsen et al. 2006), as well as three additional spectra provided by the same authors, for a gibbous, quarter and crescent moon. Quantum catches for the hawkmoth and blue tit photoreceptors were calculated under these additional light types for all scenarios (perception of flowers by hawkmoths for pollination, perception of hawkmoth wing colours by hawkmoths for signalling, and perception of hawkmoth wing colours by birds in the context of camouflage). As predicted, these changes make very little difference to our results across moonlight conditions because all natural light sources (other than sky-glow - see below) provide such broad-band spectra compared to artificial sources. This was a major undertaking (re-analysing almost all the data); all numbers, text and figures have been updated to reflect the addition of these sources.

3) Depending on the answer in number 1, you need to discuss what your assumption of using moonlight at natural nighttime lighting means for your paper. Unfortunately, I think that since you used moonlight as your only night time light reference, you are missing a big picture of the natural conditions for these moths. We know that starlight is in fact much more peaky and results in an irradiance closer to that of HPS - it would be very informative to run your models with starlight spectrum (see Optics of Life for that spectrum) and see how these ALAN spectra compare. I realize that this will be a lot of work, but you would nail this study out of the park and cover all of the bases! If you decide that including the starlight spectrum is too much work, you must discuss that you used moonlight (which represents less than 50% of nighttime lighting as the moon is below the horizon for a lot of nighttime) and thus you are only able to make comparisons of natural conditions for when there is moonlight (again, less than 50% of night). I do hope you will do the model with starlight (in which case send it to Nature or Science and ask me to be a reviewer...) as I think it will show that those red shifted anthropogenic lights are similar to starlight.

We are indeed keen to cover all bases, so have included a starlight irradiance spectrum (again from Johnsen et al. 2006, see that paper for full methods) in our analyses, for the lowest light level considered (0.001 cd.m⁻²).

As reviewer 4 suggested, we have also included several twilight spectra (corresponding to solar elevations 11.4°, 1° & -10.8° from the horizon, see Johnsen et al. 2006). All together, we now capture a much greater diversity of potential natural nocturnal illumination types, from twilight through several phases of moonlight and down to starlight, and so provide a much more comprehensive sense of hawkmoth perception in many realistic scenarios. Throughout the paper, we continue to consider the full moonlight spectrum as our standard reference at all light levels, so as to provide a consistent comparison point between artificial light types and natural illumination. However, we also include additional natural nocturnal lighting spectra as appropriate, according to natural light levels for each, and compare the performance of artificial light types to the most relevant natural illuminant at each light level. These

additional comparisons are presented in figures 2,3 & S3, and results based on different reference light types are summarised in the supplementary material (figure S9).

Small things:

1) I don't understand what you are trying to communicate in 294-296. What do you mean you only selected pairs of colours with a perfect colour match - huh? I don't understand and I know I am not giving you much to work with, wish we could talk on the phone...

Apologies if this is not clear enough – we have reworded this section. Essentially we compared the moth colours to all the thousands of background spectra under a given light type and nighttime light level e.g. LED light at 10 cd.m⁻². Then we only selected those moth- background spectra pairs which were below-threshold ($\Delta S < 1$) for colour discrimination to the hawkmoths. Then we model how these colour matches would look to a bird in the daylight. This is a test of metamerism-type effects.

We have reworded this to read (L301-304): “For each light level and light type combination, we then selected only pairs of colours that would be indiscriminable to hawkmoth vision ($\Delta S < 1$) for analysis (simulating a “perfect” background choice to hawkmoths under those viewing conditions).”

Figure S2 - what are the patches? You have numbers but I didn't see a legend for what the numbers mean. I also understand that the different species of moths have different numbers but perhaps you could say that the numbers start at one wing and then go clockwise or something like that?

These patches correspond to the numbers superimposed over the wings in Figure S1. We have edited the legend to make this clear. We have also tweaked the order of the patches in Figure S1 so that they are more logical, with the patches numbered from the base to the apex of the forewing, then the hindwing, and from the outside to the inside of any concentric marking patterns. The methods section also details how patches were selected (L228-231).

That's all. I really do think your paper is excellent and absolutely needed, I just wish starlight was in there or you discuss why you only need to think about moonlight with regards to the hawkmoth, which I believe is not the case.

I hope to see a revision,
Brett Seymoure

Reviewer #2 (Remarks to the Author):

This paper addresses the effects of various forms of artificial lighting on color perception in a moth and a bird under various intensities and contexts. It is well written, and the methods (to the extent that they can be determined -- see below) are solid. The results themselves are relatively unsurprising, that more monochromatic lights have more effect of color perception and that this effect depends on the lights' spectra and intensities, but it is good to see a systematic and comprehensive approach to this problem. I have a few concerns:

Many thanks for the positive appraisal.

1. the most major perhaps is that many of the color distances are so far above threshold that moderate to minor decreases in them should make little difference. I point the authors to this year's ProcB paper by Santiago et al that confirms what has been long assumed -- that color distances well above threshold aren't really meaningful except in certain specific contexts (e.g. attenuating media). Detection, as is typical, follows a sigmoidal curve that asymptotes once well above threshold. So I'm unsure which of the "improved" or "worse" conditions actually have meaningful effects, especially since this paper is entirely modeling without behavioral data. I am willing to believe the lighting conditions that completely destroy color perception (e.g. LPS), but this of course has been well known. There are a very large number of papers now that take a set of reflectances and illumination spectra and then calculate delta-s using the RNL model, but very few that actually groundtruth this work. Ideally, the authors should address this with behavioral trials, but at a minimum they need to find some way to tie these differences to reality, and to think carefully about which differences may truly matter.

We entirely agree that the RNL model needs to be used cautiously with suprathreshold distances. A similar point is raised by reviewers 3 and 4 below. However, even without behavioural modelling we can demonstrate that suprathreshold colour stimuli must at some critical viewing distance be exactly at threshold (where the RNL model is most appropriate). We can prove this by considering the spatiochromatic interactions between colour patches. As viewing distance is gradually increased, any two adjacent suprathreshold colours must blend together, and this degree of blending will increase beyond the receiver's contrast sensitivity such that they blend entirely. By considering spatial and chromatic factors together we can demonstrate that more dissimilar colours will be visible/detectable from larger viewing distances.

Therefore, whenever suprathreshold contrasts are measured in this study (or indeed elsewhere), an alternative (and biologically important) concept is that it will alter the maximum detectable distance, which is critical for predators detecting camouflaged prey, or moths detecting flowers. We feel that this point really requires an entire paper to discuss the effect, which has largely been overlooked by visual ecologists until recently, but we provide figure S10 as a demonstration of the distance-dependant effect of suprathreshold contrasts. In this figure we use the latest spatiochromatic modelling techniques (developed in part by Reviewer 3) to demonstrate how artificially altering the internal colour contrasts of a hawkmoth, or modelling its internal colour under different lighting conditions, will alter the maximum viewing distance at which the internal colours could be perceived.

We also clarify this spatiochromatic inference in the main text (L266 onwards):

"The RNL model is most appropriate for chromatic differences near the threshold point, however our modelling often results in suprathreshold values. Nevertheless, any two adjacent suprathreshold colours must blend to become sub-threshold at a viewing distance dependent on the receiver's acuity limits. Therefore, while our modelling may not always be ideal at close range, the suprathreshold values will scale with critical maximum detection distance, which will be larger with greater colour contrasts (we illustrate this spatiochromatic effect in fig. S10)."

2. The methods section is relatively opaque, since it mostly refers to data being pipelined

through a number of software packages such as Pavo. This is of course done in some other fields -- bioinformatics being a major offender -- but the visual ecology world is small, so it is highly unlikely that these software packages will still exist in ten year's time. Therefore it would be good to show the actual math of what is being done, since the equations will be understandable hundreds of years from now, if the last few hundred are any indication. It's also worth noting that not every researcher has access to the materials needed to run these packages, even if the packages themselves are freely available. Also, most creators of software packages provide no system that guarantees that their programs will still be available and supported in the future. The basic algorithms and math are fairly universal, but the syntax can fail when the underlying R code (for example) goes through a major update, even assuming that the program itself is safely archived. These are complex issues, but it's worth an author's time to present their methods in a form that will be usable in the future.

We regret that the reviewer finds our methods to be opaque because we have gone to considerable lengths to produce a more open and reproducible modelling and analysis workflow than the prior work in this area (also see reviewer 3's comment that the methods are well presented). Specifically, we only use open-source tools (R), and only a limited number of additional packages. Built-in functions from packages such as pavo were limited to simple uses, such as calculating cone catches for blue tit vision in daylight conditions. More complex functions are detailed in full in our code, as we were unable to use the default pavo functions for our modelling, and largely repurposed its code ourselves for low-light modelling. All this code is available as a supplementary document. Also note that these open-source tools we rely on have been deposited in GitHub or equivalent long-term repositories with version control; for example, it was possible for us to modify the "bootcoldist" function from pavo, which we use to examine differences between moth wings and natural backgrounds, as well as between flowers and vegetation, because the code underpinning it was fully available on GitHub, along with a detailed worked example of its use. We also clearly state the equations we use for the visual modelling and the values of the parameters (note that previous papers demonstrating this were difficult to follow to the point where we needed to contact the authors). We certainly hope that people will still look back at this paper in 100 years!

Reviewer #3 (Remarks to the Author):

This is a beautifully written and presented paper investigating how artificial lights may alter the perception of hawkmoth colour signals by conspecifics and potential predators. With visual modelling of spectral reflectance measurements of hawkmoths and potential backgrounds the hawkmoths could be viewed against, and the illumination spectra of artificial lights, the authors present theoretical evidence of how colour signals may be perceived differently.

Many thanks!

I have no issues with the way that the data was collected, modelled and analysed. These are very clear and presented well. However, I do have major reservations about the relevance and interpretation of chromatic contrast values calculated in the study without further behavioural validation based on reasons below.

The RNL model is commonly used to quantify chromatic contrast of colours viewed by a

given visual system under specific conditions, and under these conditions the discrimination threshold is assumed to be $\Delta S = 1$. However, the relationship between distance in colour space and perceived colour difference is not straightforward, and there has been little behavioural calibration of the model with broadband colours, such as the ones presented in this study. Recent studies that have attempted this in different types of fish have shown deviance from RNL model predictions, and that discrimination thresholds can vary between 0.7- > 3 depending on directions in colour space, see:

o Sibeaux, A., Cole, G. L. and Endler, J. A. (2019). Success of the receptor noise model in predicting colour discrimination in guppies depends upon the colours tested. *Vision Research* 159, 86-95.

o Escobar-Camacho, D., Taylor, M. A., Cheney, K. L., Green, N. F., Marshall, N. J. and Carleton, K. L. (2019). Color discrimination thresholds in a cichlid fish: *Metriacrima benetos*. *Journal of Experimental Biology* 222, jeb201160.

o Cheney, K. L., Green, N. F., Vibert, A. P., Vorobyev, M., Marshall, N. J., Osorio, D. C. and Endler, J. A. (2019). An Ishihara-style test of animal colour vision. *Journal of Experimental Biology* 222, jeb189787.

Discrimination thresholds are even higher when moving away from the neutral (achromatic) point for more highly saturated colours (which is also the case with humans).

Therefore, while it is clear that hawkmoth can perceive colour in dim light conditions, calibration of the model with behavioural experiments to show discrimination threshold are ~ 1 , has not been done (to my knowledge). It is therefore hard to interpret the measurements being obtained here, which often fall in the range of 0-2.5 (ΔS)(as per Figure 3).

This is a good point, although behavioural validation of the exact Delta-S values associated with threshold detection behaviour is well beyond the scope of this study. Nevertheless, we attempted to verify that our threshold of Delta-S=1 is reasonable, by applying our modelling to stimuli used in one paper with behavioural tests of colour vision in hawkmoths (Balkenius and Kelber, 2004), and found that our modelling was consistent with their findings. We have added this modelling to the supplementary materials (L522- 559, see fig. S11-S12). Likewise, we have repeated this for the validation of our blue tit low light visual modelling (figs S13 & S14).

However, we also note that the range of Delta-S threshold distances (~ 0.7 to ~ 3) are all around the generally accepted ranges for thresholds, and these numbers are sensible given the underlying Weber law assumptions. We can also ask how our results would be affected with different Weber fractions. We have re-run the entire analysis with larger noise-to-signal threshold ratios, and this does little to change our findings. Even more importantly, it changes our findings in a fairly linear, predictable manner. E.g. it might mean the colours switch from above-threshold to below-threshold at a slightly different light intensity. But given the extremely large log-scale of light intensities modelled the changes would be minimal.

Furthermore, 'suprathreshold' colours also have to be interpreted with caution once they go beyond the discrimination threshold. This may impact interpretation of whether colour discrimination is 'improved' or 'worse' (as per Figure 1). See recent paper: Santiago et al. (2020). Does conspicuousness scale linearly with colour distance? A test using reef fish. *Proceedings of the Royal Society B: Biological Sciences*, 287 (1935) 20201456, which addresses this issue.

Reviewer 2 raised a very similar point about suprathreshold values – please see our response, and our visual modelling demonstrating the distance-dependant spatiochromatic interactions.

Regards, Karen Cheney

Reviewer #4 (Remarks to the Author):

This paper investigates the effect of artificial light at night on colour discrimination by hawk moths. The authors address this topic thoroughly by modelling chromatic contrast under a variety of different lighting scenarios and the results suggest that some types of artificial light will hinder colour discrimination and potentially a moth's ability to background match. The paper is written very nicely and the figures are fantastic. Artificial light at night is a very important topic and any effects on insects may help us to understand declines in insect numbers. Therefore, this topic is likely to be of interest to a broad range of scientists.

Many thanks!

Before recommending this study for publication, I have some questions and comments that I think should be addressed. Most of my questions are regarding how well the models mimic environmental conditions. I am not sure that all the intensities modelled are relevant to actual conditions and some of the comparisons may not directly address the research question (detailed below). More information regarding the biology of the system would also be useful. Some of these changes may change the results and interpretation.

Comments on illumination choices:

I am interested in the choice of intensities of the lights for a few reasons. Firstly, why was moon light and D65 modelled at several different intensities? These light sources do not vary by multiple orders of magnitude in nature, so the variation isn't ecologically relevant. It would be more beneficial to keep these at natural intensity levels so the reader can see what chromatic contrast would be under natural conditions without any interference of artificial light. This would better address the research question of how artificial lights affect chromatic contrast compared to natural conditions.

Following this, it might be more useful to model dusk irradiance rather than D65 since the highest luminance of 10 cd m² is similar to dusk light levels. I would also be interested to see starlight irradiance since you mention that these moths can discriminate colours under starlight conditions.

We agree that we could use more relevant light sources at different modelled intensities, although natural sources such as moonlight vary across multiple orders of magnitude with only minimal changes in spectral curve shapes (e.g. phase and atmospheric conditions). Nevertheless, reviewer 1 also suggested the inclusion of starlight and additional moonlight spectra in our analyses to increase the relevance of our results to more nocturnal lighting conditions, and we have now included irradiance spectra for different moon phases (full, gibbous, quarter and crescent), as well as

several twilight spectra (corresponding to solar elevations 11.4°, 1° & -10.8° from the horizon, see Johnsen et al. 2006), and starlight conditions. As detailed in the response to reviewer 1, we decided to keep moonlight and D65 spectra in our analyses at different light levels, to provide a consistent reference point against which we can compare the performance of each artificial light type. However, we recognise that it would also be relevant to understand how artificial light types compare to the most appropriate natural lighting conditions at each light level (see figures 2,3, & S3). We have therefore added relevant natural light spectra to our analyses at each light level and performed additional analyses, comparing our artificial light types to a more appropriate natural light type at each light level. The results of these analyses are presented in the supplementary material (figure S9).

It would also be great to have some more information on the intensity of different artificial light sources. I imagine they are not all the same intensity in the field – do they vary greatly?

Certainly they do vary, but artificial light sources are in practice extremely limited in their upper intensity. Note the log-scale over which light intensities vary, increasing an artificial light intensity ten-fold would generally be uneconomical (e.g. such intense night-time light is only typically associated with sports stadia). The lower intensity has no limit, and given intensity will fall off at $1/r^2$, the intensity of artificial lights decreases dramatically with distance.

Furthermore, the lowest luminance measurement corresponds to low level skyglow. Therefore, it is relevant to model that for specific light sources when skyglow is likely a mix of multiple light sources? It may be more relevant to obtain a measure of skyglow (potentially with and without moonlight) for these models. Would moonlight mix with artificial light sources at other intensities also?

Yes, absolutely (see also reviewer 1's comments) – we have included a starlight/skyglow mixture.

Other comments:

I am also curious as to why the von Kries transformation was not applied. A previous study investigated contrast for these moths with and without the von Kries transformation (Johnsen et al 2006), and Kelber et al 2003 behaviourally demonstrated colour constancy. This would likely have a large effect on the results. Sorry if I have misunderstood!

This is a great question, which highlights a benefit of our customised low-light RNL chromaticity space. This is a Fechnerian (i.e. perceptually uniform) colour space, meaning that if we were to apply the von Kries transform it would shift the achromatic point to the centre of the space, and would shift all other measurements by an identical vector. The result is that any pairwise distances between any two colours in this space are completely independent of where exactly the achromatic point is, so applying the von Kries would make no mathematical difference to any of our analyses. Euclidean distances in this space are identical to ΔS values. Note that if we had attempted to measure saturation/chroma (distance from the achromatic point) or hue (angle from the achromatic point), we would have needed to apply the von Kries transform. This feature of our colour space is particularly convenient for

low-light modelling in nature, where we can never be certain where exactly the achromatic point is and how it might be limited by photon shot-noise.

Johnsen et al 2006 found an effect of applying the von Kries transform because they used a non-Fechnerian colour space (the Maxwell triangle), which does not shift colour coordinates by an identical vector. We have discussed these modelling differences with Sonke Johnsen, and are confident our approach is best practice.

More natural history about the system would also be useful to understand the conditions selected in the model. For example, are they often found near the lights or right under the lights?

*We do find hawkmoths (including elephant hawkmoths) attracted to flowers and moth traps in residential gardens in Penryn/Falmouth which are illuminated by streetlights, and on our University campus, which has ALAN levels typical of a built area. We have also found larvae on *Epilobium* spp. Growing in walls directly below streetlights, presumably from eggs laid in situ. However, given the sensitivity of their vision to low light levels, light pollution will be visible to the moths across most of their European range.*

Moth trapping is generally more successful on darker nights (low natural illumination) and in areas with lower levels of ALAN; possibly due to higher population levels, but also likely due to the higher effectiveness of traps to attract moths from larger distances with lower ALAN. Moth trapping under the influence of ALAN can therefore not be used to reliably assess population levels beyond presence/absence.

We have clarified this in L57-60: "Nocturnal moths such as hawkmoths (Sphingidae) have low-light colour vision which allows them to locate flowers even under starlight levels of illumination^{5,23,25}, meaning much of their natural range will be subject to visible light pollution, and they are found in built areas near streetlights."

Are the flowers often found near/under lights?

Yes, indeed we were able to collect almost all the flower species required from our University campus or from residential gardens in Penryn.

Do they fly at dusk or only night time?

Elephant hawkmoths are generally considered fully nocturnal, however behaviour varies among moth (and other hawkmoth) species, and is often not well known (and in practice probably a lot more varied than simple diurnal/crepuscular/nocturnal classifications). It likely depends on individual condition and environmental factors such as temperature and light levels. Nevertheless, our modelling covers the entire range of dusk and nocturnal lighting conditions.

It would be great if you could add in a paragraph discussing achromatic discrimination. I am curious whether the moths could compensate for any loss in colour discrimination via achromatic discrimination.

We were keen to look at this issue, which should be affected by ALAN, but simply do not have enormous confidence in current models even before we would have to adapt them for low-light vision. Until recently it was considered a less complicated area of animal visual modelling than chromatic differences, however recent research

in animals such as van den Berg et al. 2020, JEB doi:10.1242/jeb.232090 (and a vast literature in humans) points out how fraught this topic is, and how current modelling practices are probably inadequate. Nevertheless, chromatic contrasts alone are known to be key for the aspects of visual ecology which we explore here, particularly in a signalling context such as flowers attracting pollinators. This is because achromatic channels inevitably have much higher levels of “noise” in a visual scene than chromatic channels.

I like the logic behind your models of background matching! However, I'm unsure about the interpretation. It looks like the moths don't match the background, even if they select their background under moonlight conditions. Chromatic contrast may not be linear at these high JNDs, so the increase may not be meaningful (c.f. Santiago et al 2020, Proc R Soc B). So the artificial lights might not make it much worse.

Absolutely, a point echoed by reviewers 2 and 3. Please see our response above concerning the need to consider spatiochromatic properties of visual modelling, and how there must be a (biologically important) critical viewing distance where the colours are at the threshold.

Minor comments:

Is this study relevant to other nocturnal/crepuscular organisms? Are there others that use colour vision?

It is likely that our findings are relevant to a large number of other visually-guided nocturnal species such as moths. However, this is currently somewhat speculative because so little is known about the visual systems of other moths (e.g. two noctuid species have been shown to have colour vision). Otherwise colour vision under very low levels of illumination is still comparatively rare in terrestrial animals (e.g. found in some geckos and amphibians). Nevertheless, our modelling is relevant for almost all visually-guided species whose diurnal (photopic) colour vision will deteriorate at lower light levels (the switch to mesopic/scotopic levels). Given the importance of dawn and dusk for the activities of many crepuscular and diurnal animals, modelling the effects of ALAN could be interesting far beyond moths. We hope that by drawing attention to these issues we will encourage more research into this area.

Are each of the light sources chosen commonly found in the human modified environments?

We selected our light sources to represent the main sources used as street lights across Europe in the last few decades. LEDs of different CCTs make up the majority of new installations (hence we selected two of the largest manufacturers and a wide range of CCTs). Mercury vapour sources are now banned in most European countries, but were formerly widespread in some countries, and are still frequently used for moth trapping. High and Low-pressure sodium were widespread, are still frequently used but often phased out in favour of LEDs. PC-Amber LEDs are rare but have been proposed as a ecologically friendly / light pollution friendly alternative. Narrowband amber LEDs are not used, however these theoretically offer an interesting modern alternative to LPS, and for the purposes of our study offer an interesting comparison to the broadband PC-amber (same peak wavelength, but different distribution).

Line Comments:

Line 30: Any references for sexual signalling? Maybe fireflies?

Many thanks for the suggestion, we have included a citation to a recent review on the threats to fireflies.

Figure 1: to take up less space you could split the legend into two columns to fill up the white space to the right. Also, it looks like only 7 species in each cell for b&c? Really great figure!

Thank you, and we have moved the legend into two columns to save space, as suggested. For columns b & c, all 14 species are represented (in the order shown in the "Viewing context" row) - we have added thin dashed lines to separate the light levels, to make it easier to see which group of cells correspond to each column, and so more clearly show that all 14 species are present.

Line 63: Was the background foliage a combination of many plants?

Yes, including leaves from the same plant as the petals, but also a range of different grass types (because many flowers are so spatially removed from their plant's leaves). The methods section details the exact foliage spectra used in the analysis (L62-66, and L230-238)

Line 217-220: Are the colours structural? If they are pigment based, do they match alive hawk moths? Some are quite old and the pigments might have started to break down.

We assume that at least some of the colours are structural given structural colours are common in Lepidoptera (particularly the blue eye spots on eyed hawkmoths). Nevertheless, we used a range of museum specimens, and always the most recent wings available (we show the collection dates in table S2). We also had a fresh sample of elephant hawkmoth wings (the most colourful species), which was no brighter in colour than the museum specimens of the same species.

Line 250: What intensities are relevant to artificial lights? I see it is mentioned in the supp info (Lines 396-298. I think this needs to be included in the main text to provide context.

We have clarified that "Typical artificial streetlight intensities also cover this range [10-0.001 cd.m⁻²] dependent on distance to the light." The ground-level intensity near street lights is variable, but typically near 1. Higher values are only typical nearer the source (e.g. vegetation raised above the ground). The lower values are all plausible at specific distances and atmospheric conditions.

Line 284-299: It would be good if you could briefly restate the logic here (i.e. a moth chooses where to sit at night, but needs to match throughout the day for diurnal predators).

We have re-written this paragraph, and hope that it is more clear. L297 onwards now reads:

"Finally, moths must choose where to rest at night (potentially under the influence of artificial light) but rely on a day-time colour match to their background to protect them from predators. To quantify this effect we calculated chromatic contrasts between every moth forewing colour (NFOREWING = 235) and natural background colour

(NBGD = 953) for hawkmoth vision in low light as described above, under each combination of light type and light level. For each light level and light type combination, we then selected only pairs of colours that would be indiscriminable to hawkmoth vision ($\Delta S < 1$) for analysis (simulating a “perfect” background choice to hawkmoths under those viewing conditions).”

I hope you find this helpful.

REVIEWER COMMENTS

Reviewer #1 (Remarks to the Author):

I appreciate the authors working so diligently on all of the comments from us reviewers. I do think that the paper has been improved to a level where it will make a big impact across disciplinary approaches to artificial light at night. I am really glad to see the authors include starlight, although I still have a few remaining questions pertaining to starlight. Why isn't it included in figure 1? Why isn't starlight compared to moonlight? These are very different spectrally speaking. I do think more could be discussed about the findings relative to the natural light spectra (moon and starlight), but I do understand that you are constrained by word limit. I hope COVID goes away and soon we can share a pint and talk all about this at a meeting. I really like this study, nice work! - Brett Seymoure

Reviewer #2 (Remarks to the Author):

I am satisfied with both the revisions and the responses to my concerns (and those of the other reviewers). Nice work!

Reviewer #3 (Remarks to the Author):

I think the authors have done an excellent job of revising the manuscript.

My only remaining reservation is that I feel the manuscript still implies a 'hard' threshold of $\Delta S = 1$: anything over this is discriminable and anything under is not. This is most notable in the figures. As we are beginning to collect more behavioural data from birds, fish and bees, it is clear that behavioural thresholds vary widely depending on the hue/saturated tested and viewing conditions, which are not explained well by the RNL model. Therefore, I think the visual ecology field should move away from this rigid view of $\Delta S = 1$ as the threshold, and this paper could highlight that we need to take a more considered and conservative approach when modelling data against discrimination thresholds (and act as an exemplar for future studies).

The easiest way of addressing this issue is to add a coloured band to highlight uncertainty rather than a straight line for the threshold lines on Figures 2 and 3. This could be between 0.5-1.5 for bird vision, for example, or even larger/more conservative for the hawkmoth, for which we have less behavioural data. They could also be labelled theoretical thresholds or putative thresholds to make this point clear.

Reviewer #4 (Remarks to the Author):

First off, I would like to thank the authors for their thoughtful and detailed response to all the reviewer's comments. Everything was explained really well, and substantial effort was put into reanalysing the data to respond to the comments – Thank you! It is a very well written and thorough study and the figures are fantastic.

The main comment I have is again in regards to the intensities of lights and how this relates to environmental conditions. I think it's relatively easy to run visual models, but it's essential to ensure that the models mimic environmental conditions as much as possible. My concern with the manuscript at the moment is that the models and comparisons do not mimic environmental lighting conditions adequately.

Comparisons within intensities: Wouldn't two light sources of the same intensity mix together? Therefore, some of the lights might not be as bad as they appear for chromatic contrast because the natural source of light could broaden the spectrum. If comparing within an intensity, I would have thought that to model an artificial light, mixing of the artificial light with natural light is always relevant. I am interested to hear the authors thoughts on this (have I misunderstood something?). For me, the comparisons across intensities are more realistic (see below), which would require rejigging the focus of the manuscript.

Comparisons across intensities: When the artificial light source is higher intensity than the natural light (e.g. if the moth was directly under an artificial light when the natural light is starlight), one light source would dominate and the results would likely be driven by the dominant light source (i.e. more similar to what you've presented). In this case, it would be interesting to compare across intensities rather than within intensities. For example, if it's a really low light night (no moon for example), then moths near to many of the lights will be more easily able to discriminate colours than natural conditions. How might these patches of improved discrimination affect the moths?

On a different note, more discussion about how the implications of your results compare to other studies would be interesting. Bioluminescent signals (marine or terrestrial) may benefit from a dark background or narrow band lights. Your results show that this is not necessarily the same for moths. Expansion of the final paragraph to further discuss the conclusions of your study in comparison to other studies would be great for providing broader implications.

Lines 189-198: Thanks for addressing mine and other reviewers' comments on this part. I think it is important to highlight here that even under moonlight the chromatic contrasts are relatively high, and mention the spatiochromatic inference here, rather than in the methods. From the additional modelling that you added (good work – I really like it!), it doesn't look like the increase in ΔS will necessarily have a huge effect on viewing distance.

I realise you are probably limited on space, but I do think it would really benefit the manuscript.

Minor Comments (feel free to ignore if you disagree):

Line 10: Is PC amber LED lighting the commonly used name? LED is a well known acronym but I don't know what the PC means. Could write out what PC means instead of the acronym.

Line13-16: This sentence is a bit convoluted, but it is tricky to think of clearer wording. Could split into two sentences or maybe streamline the sentence a little more (e.g could remove "in environmentally sensitive areas").

Line 90: "Thresholds for)" Thresholds for "e"?"

Figure 2: I didn't notice the moth in the schematic last time – I love it! Very helpful.

Line 354: "Figure"

Line 390: Caption and aspects in the figure say "relative to moonlight" except this figure is relative to different reference illuminants.

We thank the reviewers for their continued enthusiasm for our study and appreciation of our revisions. We have responded to their further suggestions below (in *blue italics*).

REVIEWER COMMENTS

Reviewer #1 (Remarks to the Author):

I appreciate the authors working so diligently on all of the comments from us reviewers. I do think that the paper has been improved to a level where it will make a big impact across disciplinary approaches to artificial light at night.

Many thanks for the positive appraisal.

I am really glad to see the authors include starlight, although I still have a few remaining questions pertaining to starlight. Why isn't it included in figure 1? Why isn't starlight compared to moonlight? These are very different spectrally speaking.

Our reanalysis does now include starlight, although we have not included it in Figure 1; this is partly because it does not change any of the effects of artificial light at respective light intensities; and partly in order to keep the figures accessible (there are many other potential combinations of light sources we could compare to). While the spectrum of starlight (actually a combination of starlight and sky-glow) is very different to moonlight, it does not actually behave differently in terms of moth vision. Despite the peaks, it is still a broad-spectrum light source, and at the extremely low light levels where it occurs it actually behaves similarly to other broad-spectrum sources to moth vision. The effects of starlight can be determined from other figures (e.g. fig. 2, where D65, full-moon and starlight spectra produce equivalent chromatic contrasts).

I do think more could be discussed about the findings relative to the natural light spectra (moon and starlight), but I do understand that you are constrained by word limit.

It would indeed be great to explore these ideas more. In this study we want to focus on the anthropogenic issues surrounding artificial light sources. However, see below about our desire to expand this modelling to consider real-world measured habitat mosaic lighting effects, which would include both artificial and natural light comparisons.

I hope COVID goes away and soon we can share a pint and talk all about this at a meeting. I really like this study, nice work! - Brett Seymoure

Likewise!

Reviewer #2 (Remarks to the Author):

I am satisfied with both the revisions and the responses to my concerns (and those of the other reviewers). Nice work!

Many thanks!

Reviewer #3 (Remarks to the Author):

I think the authors have done an excellent job of revising the manuscript.

Many thanks!

My only remaining reservation is that I feel the manuscript still implies a 'hard' threshold of $\Delta S = 1$: anything over this is discriminable and anything under is not. This is most notable in the figures. As we are beginning to collect more behavioural data from birds, fish and bees, it is clear that behavioural thresholds vary widely depending on the hue/saturated tested and viewing conditions, which are not explained well by the RNL model. Therefore, I think the visual ecology field should move away from this rigid view of $\Delta = 1$ as the threshold, and this paper could highlight that we need to take a more considered and conservative approach when modelling data against discrimination thresholds (and act as an exemplar for future studies).

The easiest way of addressing this issue is to add a coloured band to highlight uncertainty rather than a straight line for the threshold lines on Figures 2 and 3. This could be between 0.5-1.5 for bird vision, for example, or even larger/more conservative for the hawkmoth, for which we have less behavioural data. They could also be labelled theoretical thresholds or putative thresholds to make this point clear.

We agree entirely (and welcome your own work in testing the validity or otherwise of these models). Ultimately we believe the models should be improved to account for these effects, but in the meantime we have altered figures 2 and 3 as you suggested, so that there is no longer a hard threshold at 1. We have also altered the legends of figures 1 and S9, and made changes to the manuscript making it clear that this is an approximate threshold. Note, however that we have: i) justified these threshold values by testing our model against behavioural data in moths and blue tits, ii) re-run our modelling with different Weber fractions, and iii) point out the spatiochromatic effects. So we believe we are acting as a good example for the field.

Reviewer #4 (Remarks to the Author):

First off, I would like to thank the authors for their thoughtful and detailed response to all the reviewer's comments. Everything was explained really well, and substantial effort was put into reanalysing the data to respond to the comments – Thank you! It is a very well written and thorough study and the figures are fantastic.

Many thanks for the positive appraisal.

The main comment I have is again in regards to the intensities of lights and how this relates to environmental conditions. I think it's relatively easy to run visual models, but it's essential to ensure that the models mimic environmental conditions as much as possible. My concern with the manuscript at the moment is that the models and comparisons do not mimic environmental lighting conditions adequately.

We respond to your specific concerns below.

Comparisons within intensities: Wouldn't two light sources of the same intensity mix together? Therefore, some of the lights might not be as bad as they appear for chromatic contrast because the natural source of light could broaden the spectrum. If comparing within an intensity, I would have thought that to model an artificial light, mixing of the artificial light with natural light is always relevant. I am interested to hear the authors thoughts on this (have I misunderstood something?). For me, the comparisons across intensities are more realistic (see below), which would require rejigging the focus of the manuscript.

Two light sources of the same intensity would indeed mix together, and it would – in principle – be straightforward to perform our modelling on the resulting spectrum. In real-world conditions however, the point where these light sources occur at equal intensities is exceedingly small. Note, for example, how our modelling occurs over five orders of magnitude, and how light intensities in general follow a log-normal distribution. Therefore the ecological relevance of blending equal intensities of lighting is highly questionable.

Therefore we believe future work should measure the real-world habitat mosaic effects of natural and artificial light at night, creating an ecologically relevant dataset for modelling.

Comparisons across intensities: When the artificial light source is higher intensity than the natural light (e.g. if the moth was directly under an artificial light when the natural light is starlight), one light source would dominate and the results would likely be driven by the dominant light source (i.e. more similar to what you've presented). In this case, it would be interesting to compare across intensities rather than within intensities. For example, if it's a really low light night (no moon for example), then moths near to many of the lights will be more easily able to discriminate colours than natural conditions. How might these patches of improved discrimination affect the moths?

Indeed, our study already models the effects of switching between light sources at different intensities. Notwithstanding the above problems of blending of spectra (which under log-scale intensities will often have a small effect in natural conditions), our rationale has explicitly assumed that a moth will be moving between patches of different intensities and spectral distributions. A reader can readily interpret the effects of a moth moving between different intensities from the figures (and data in the supplementary materials). Taking your example and looking at figure 2, according to our modelling a moth would be able to discriminate a flower from its background under moonlight at almost all intensities (except perhaps pink/purple flowers at 0.001 cd.m^{-2} , which are borderline), however if that moth moved to an area of PC amber light it would only be able to discriminate pink/purple flowers at intensities above 0.1 cd.m^{-2} , but even then not quite as well under natural light (the small amount of blended moonlight wouldn't overcome the dominance of medium/longwave light overwhelming the UV and shortwave detection). The detection of pink/purple flowers would only become as good as moonlight at extremely high intensities of PC amber (10 cd.m^{-2}) within a few metres of the bulb itself, where any effect of blending with natural light will become irrelevant.

Regarding how patches of improved discrimination might affect the moths, we do find conditions where artificial sources (e.g. mercury vapour and pink/purple flowers) are estimated to create larger chromatic differences than natural light, however we are reluctant to be drawn on speculation here due to a lack of behavioural evidence and word limits.

On a different note, more discussion about how the implications of your results compare to other studies would be interesting. Bioluminescent signals (marine or terrestrial) may benefit from a dark background or narrow band lights. Your results show that this is not necessarily the same for moths. Expansion of the final paragraph to further discuss the conclusions of your study in comparison to other studies would be great for providing broader implications.

We welcome this suggestion and have re-written the final paragraph substantially to contrast our results with those suggesting amber lighting is less harmful to wildlife in general, and to fireflies specifically. This highlights the fact that there is unlikely to be "one-size-fits-all" solution to ALAN mitigation.

Lines 189-198: Thanks for addressing mine and other reviewers' comments on this part. I think it is important to highlight here that even under moonlight the chromatic contrasts are relatively high, and mention the spatiochromatic inference here, rather than in the methods. From the additional modelling that you added (good work – I really like it!), it doesn't look like the increase in ΔS will necessarily have a huge effect on viewing distance. I realise you are probably limited on space, but I do think it would really benefit the manuscript.

We are glad you appreciate the spatiochromatic modelling, which is relevant to all of the modelling work in our study. We have considered describing this effect in the main text, but consider it too technical and disruptive to the main discussion, so would rather keep it in the methods section of the main text.

We believe this shows a potentially large effect though. i.e. the moth's internal colours would only be visible at 10cm with the white LED and reduced chromaticity (implying either a reduced chromatic sensitivity in the receiver or more muted colours in the moth itself), while under all other light conditions the same colours would be visible from 20cm. Now consider the moths are searching for the target in a three-dimensional space; a sphere of 10cm radius has a volume of $\sim 4,200\text{cm}^3$, while a sphere of 20cm radius has a volume of $\sim 33,500\text{cm}^3$. Ultimately this means a comparatively subtle change in lighting or patterning can make the volume of space in which a signal is detectable roughly 8 times larger.

Minor Comments (feel free to ignore if you disagree):

Line 10: Is PC amber LED lighting the commonly used name? LED is a well known acronym but I don't know what the PC means. Could write out what PC means instead of the acronym.

PC amber LEDs are phosphor-converted LEDs – we have added in the full name in the abstract, and in the main text when this light type is first mentioned.

Line13-16: This sentence is a bit convoluted, but it is tricky to think of clearer wording. Could split into two sentences or maybe streamline the sentence a little more (e.g could remove “in environmentally sensitive areas”.

We have made a small change to hopefully make this sentence more readable.

Line 90: “Thresholds for)” Thresholds for “e)”?

Corrected, thank you.

Figure 2: I didn't notice the moth in the schematic last time – I love it! Very helpful.

Thanks!

Line 354: “Figure”

Fixed, thank you.

Line 390: Caption and aspects in the figure say “relative to moonlight” except this figure is relative to different reference illuminants.

We have corrected this in the figure and figure legend.

REVIEWERS' COMMENTS

Reviewer #4 (Remarks to the Author):

Thanks again for your detailed and clear responses. I appreciate the time you have put in to responding to the comments. I have no more comments and look forward to seeing it in print. Great work.